# Bibliometric Analysis of Hotspots and Frontiers of Immunotherapy in Pancreatic Cancer

**DOI:** 10.3390/healthcare11030304

**Published:** 2023-01-19

**Authors:** Qiong Xu, Yan Zhou, Heng Zhang, Haipeng Li, Haoren Qin, Hui Wang

**Affiliations:** 1School of Integrative Medicine, Tianjin University of Traditional Chinese Medicine, Tianjin 300193, China; 2Department of Oncology, Institute of Integrative Oncology, Tianjin Union Medical Center, Nankai University School of Medicine, Tianjin 300071, China; 3Clinical College of Neurology, Neurosurgery and Neurorehabilitation, Tianjin Medical University, Tianjin 301700, China

**Keywords:** pancreatic cancer, immunotherapy, bibliometric analysis, VOSviewer, CiteSpace

## Abstract

Background: Pancreatic cancer is one of the most common malignant neoplasms with an increasing incidence, low rate of early diagnosis, and high degree of malignancy. In recent years, immunotherapy has made remarkable achievements in various cancer types including pancreatic cancer, due to the long-lasting antitumor responses elicited in the human body. Immunotherapy mainly relies on mobilizing the host’s natural defense mechanisms to regulate the body state and exert anti-tumor effects. However, no bibliometric research about pancreatic cancer immunotherapy has been reported to date. This study aimed to assess research trends and offer possible new research directions in pancreatic cancer immunotherapy. Methods: The articles and reviews related to pancreatic cancer immunotherapy were collected from the Web of Science Core Collection. CiteSpace, VOSviewer, and an online platform, and were used to analyze co-authorship, citation, co-citation, and co-occurrence of terms retrieved from the literature highlighting the scientific advances in pancreatic cancer immunotherapy. Results: We collected 2475 publications and the number of articles was growing year by year. The United States had a strong presence worldwide with the most articles. The most contributing institution was Johns Hopkins University (103 papers). EM Jaffee was the most productive researcher with 43 papers, and L Zheng and RH Vonderheide ranked second and third, with 34 and 29 papers, respectively. All the keywords were grouped into four clusters: “immunotherapy”, “clinical treatment study”, “tumor immune cell expression”, “tumor microenvironment”. In the light of promising hotspots, keywords with recent citation bursts can be summarized into four aspects: immune microenvironment, adaptive immunotherapy, immunotherapy combinations, and molecular and gene therapy. Conclusions: In recent decades, immunotherapy showed great promise for many cancer types, so various immunotherapy approaches have been introduced to treat pancreatic cancer. Understanding the mechanisms of immunosuppressive microenvironment, eliminating immune suppression and blocking immune checkpoints, and combining traditional treatments will be hotspots for future research.

## 1. Introduction

Pancreatic ductal adenocarcinoma (PDAC) is a common malignant tumor of the digestive system and one of the leading causes of cancer-related deaths today, and is principally seen in men and in advanced age groups (40–85 years) [1]. Although there are many causes for pancreatic cancer, smoking and family history appear to be dominant factors [2]. It is reported that about 5~10% of pancreatic cancers are caused by genetic factors [3]. NCCN guidelines suggested the mutational analysis of genes involved in syndromes associated with PDAC onset. There are many types of genetic mutations associated with hereditary pancreatic cancer, including ATM, BRCA1, BRCA2, CDKN2A, PALB2, STK11, MLH, MSH2, MSH6, and PMS2. Among them, BRCA2 mutations are the most common genetic alteration, observed in about 5–10% of cases [4]. For the treatment of patients with BRCA mutations, NCCN guidelines recommend the combination of gemcitabine (GEM) and cisplatin (CDDP) [5]. PARP inhibitors have been clinically investigated for BRCA mutated cases [6]. Despite advances in surgery, chemotherapy, and radiotherapy, mortality rates remain high. Many patients are already in advanced stages at the time of diagnosis [2,7]. In this case, the focus of treatment has shifted to improving overall survival using immunotherapy, targeted therapy, and endocrine therapy. A clinical trial of anti-programmed cell death protein 1 (PD-1) antibody reported a positive response in PDAC patients with MMR alterations [4]. Thus, among those strategies, immunotherapy may be one of the most promising.

Immunotherapy can boost natural defense to eliminate malignant tumor cells and it is a monumental breakthrough for cancer treatment [8]. Several clinical studies have shown immunotherapy works continuously on tumor cells [9,10,11,12]. The major categories of immunotherapy include immune checkpoint inhibitors (ICIs), cancer vaccines, cytokine therapies, adoptive cell transfer, and oncolytic virus therapies. ICIs have been applied in the treatments of many cancer types, including melanoma, lung cancer, and renal-cell cancer [13]. In healthy individuals, the immune checkpoint regulates the actions of ligands and receptors to keep the immune functions of T cells normal and balanced. When T cells are activated, they will express more immune checkpoint receptors, such as PD-1 or cytotoxic T lymphocyte-associated antigen 4 (CTLA-4) [14,15]. In 2007, programmed cell death-ligand 1 (PD-L1) was first identified as a new pancreatic cancer prognostic indicator due to the confirmation of up-regulation of PD-L1 expression in human pancreatic cancer cell [16]. Anti-CTLA-4 and anti-PD-1/anti-PD-L1 agents can inhibit immune checkpoints, and make T cells activate and provide effective approaches for tumor immunotherapy [17]. However, several immunotherapies that have been successful in various types of cancer are less effective on pancreatic cancer due to its low tumor mutation burden and immunosuppressive tumor microenvironment [18,19]. Much research has indicated that pancreatic cancer may evade the immune surveillance by inducing the development of immune-suppressive T cells, such as regulatory T (Treg) cells. Treg cells exert immunosuppressive functions through multiple mechanisms [20]. One mouse experiment of pancreatic cancer confirmed that the combination PD-1 inhibitors and OX40 agonists reduced the proportion of Treg and exhausted T cells, and increased the number of CD4+ and CD8+ T cells in pancreatic tumors [21]. Thus, an increasing amount of research on immunotherapy has been conducted in recent years, and the research tends to overcome immune suppression and combine immunotherapy with traditional strategies to treat pancreatic cancer in the future.

Medical big data and data mining is used to analyze a large amount of medical data and dig out valuable diagnostic rules to provide reference for the diagnosis and treatment of diseases. Bibliometrics is the analysis of a large amount of data to identify research hotspots and explore cutting-edge trends in recent years by revealing the structure of knowledge in a scientific field over a certain period of time [22,23]. However, the global bibliometric analysis on the knowledge-mapping of immunotherapy for pancreatic cancer has not yet been performed.

This study is based on immune-related studies of pancreatic cancer included in the Web of Science Core Collection, and we used visualization analysis software CiteSpace and VOSviewer to generate a scientific knowledge map. Through co-occurrence and cluster analysis of related literature research institutions, keywords, and authors, among others, we explored the development hotspots and frontier trends of pancreatic cancer immunology research, and provide support and reference for the subsequent related research.

## 2. Materials and Methods

### 2.1. Dataset Selection

The Science Citation Index Expanded of Clarivate Analytics’ Web of Science Core Collection (WoSCC) was used as the data source. The WoSCC database is regarded as one of the most comprehensive, systematic, and authoritative databases, and contains over 15,000 influential high-quality journals from all over the world. According to previous studies, papers in the Web of Science Core Collection (WOSCC) can represent the status of medical science; therefore, we chose WOSCC as our data source. WoSCC has been widely used for bibliometrics analysis and visualization of scientific literature in a substantial number of studies [24,25,26,27]. We selected articles about pancreatic cancer immunotherapy on 10 March 2022. The main search terms were as follows: #1, “Immunotherap*” OR “Anti-CTLA-4” OR “Anti-PD-1” OR “Anti-PD-L1” OR “Ipilimumab” OR “Tremelimumab” OR “Pembrolizumab” OR “Atezolizumab” OR “GVAX” OR “Algenpantucel-L” OR “Dendritic cell vaccine” OR “K-Ras vaccine” OR “CAR T cell therapy” OR “GV1001 vaccine”; #2, “Pancreatic cancer*” OR “Pancreatic carcinoma” OR “Carcinoma of pancreas” OR “Cancer of pancreas” OR “Pancreas neoplasm*” OR “Pancreas cancer*”; #3, “#1” AND “#2”.

### 2.2. Article Selection

The researchers screened all articles by the title and abstract and selected the article that fits the topic. Data was downloaded and analyzed independently by two researchers to ensure its accuracy and reliability. When the two researchers had different opinions regarding the data, a third researcher decided the prevailing opinion. The exclusion criteria were document types (meeting abstracts, editorial materials, corrections, letters, news items, proceedings papers, book chapters, or early access). The publications from January 2007 to December 2021 were searched. The language was limited to English, excluding German, French, Polish, Arabic, Czech, and Estonian. The literature types were limited to “articles” and “review articles”.

### 2.3. Data Visualization and Analysis

We used VOS viewer (version 1.6.18) [23], Citespace (version 5.8.R3) [22], and an online platform for bibliometric analysis (http://bibliometric.com (accessed on 10 March 2022)) to construct visualization networks between researchers, journals, institutions and countries. Co-authorship, citation, co-citation, and co-occurrence analysis were also achieved. The data analysis process strictly followed the corresponding statistical process. In VOSviewer, nodes represent different objects, such as countries/regions, institutions, and researchers. The circle size of nodes represented frequency. Meanwhile, CiteSpace was used for the co-citation analysis of references and the extraction of keywords and references with high citation bursts. In addition, CiteSpace generated a dual-map overlay of journals. The setting parameters were as follows: top 30 for selection criteria, minimum spanning tree and pruning sliced networks for pruning, and one or two years per slice from 2007 to 2021. In addition, the visualization of global publications and citation trends was analyzed by the online citation report of WoS.

## 3. Results

### 3.1. Trend of Global Publications and Citations

In the first stage, 2812 papers were searched out. After excluding non-English and irrelevant articles, a total of 2475 documents (1702 articles and 773 reviews) were included in the study (Figure 1).

According to Figure 2A, we can see that the number of papers gradually increased from 34 in 2007 to 529 in 2021. From 2014, more than 100 articles were published every year. In the last five years, this area has developed rapidly and published 1721 (69.54% of 2475) articles. There were 91,208 citations for all papers, with an average of 36.85 citations per paper.

### 3.2. Country/Region and Institution Analyses

A total of 72 countries/regions had published articles on pancreatic cancer immunotherapy (Figure 2B). Figure 2C shows the trend in the number of articles from the top 10 countries. The United States (US) was the most productive country, with 1096 publications (Table 1), followed by China (567) and Germany (222). Additionally, the H-index of the US far exceeded that of China. Although China was ranked as the second most productive country, the average number of citations per article (18.63) was lower than those for Germany (37.35), Japan (33.97), and Italy (43.33), indicating a relatively low impact of Chinese publications and a lack of high-quality publications. Furthermore, the collaboration analysis according to country/region revealed that the US collaborated with many countries, including China and Italy. In contrast, collaboration among other countries was weak (Figure 2D). The citation network map in Figure 2E shows the citation relationships among the top 30 countries/regions in terms of the number of articles.

A total of 2880 institutions contributed to the field of pancreatic cancer immunotherapy, with the US accounting for 7 out of 10 institutions (Table 2). Johns Hopkins University had the highest contribution, with 103 articles; the second ranking institution was the University of Texas MD Anderson Cancer Center (80 articles), followed by the University of Pennsylvania (70 articles). In the collaboration analysis according to institution (Figure 3A), we only selected institutions with 10 or more articles; accordingly, 141 institutions were analyzed. The collaboration analysis organized these institutions into 10 clusters, with especially close collaboration within each cluster. The institution citation analysis revealed that the top three institutions in terms of the total link strength (TLS) were the University of Pennsylvania (TLS = 2384), Johns Hopkins University (TLS = 2313), and University of Texas MD Anderson Cancer Center (TLS = 1533) (Figure 3B, Table 2).

### 3.3. Authors and Co-Cited Authors

In total, 14,998 authors and 59,839 co-cited authors were identified in this study. Table 3 shows the 10 most prolific authors and the top ten co-cited authors with the largest TLS. EM Jaffee was the most productive author, with 43 articles, followed by L Zheng and RH Vonderheide, with 34 and 29 articles, respectively. The collaboration analysis revealed that authors generally did not collaborate closely (Figure 4A). EM Jaffee and L Zheng were placed at central positions in the clusters, and collaborated closely with each other. However, L Zheng and RH Vonderheide lacked collaboration. In the co-citation analysis, there were 5 clusters, 277 nodes, and 33,190 links in the network map (Figure 4B). DT Le had the highest centrality, and had close contact with other scholars. The top three authors in terms of the TLS were DT Le (TLS = 20,853), GL Beatty (TLS = 17,327), and JR Brahmer (TLS = 10,346).

### 3.4. Contributions of Top Journals

A total of 612 academic journals published articles in the field of pancreatic cancer immunotherapy from 2007 to 2021, including 111 journals with more than 5 articles. Cancers was the most active journal, with 99 articles in the field; however, its impact factor (IF) was much lower than that of the second-ranked journal, Clinical Cancer Research (6.639 vs. 12.53). The journal with the third highest number of publications was Oncoimmunology (69 articles, IF = 8.11). The Journal for Immunotherapy of cancer had the largest IF (13.75), followed by Clinical Cancer Research (12.53) and Cancer Research (12.701). The majority of the top ten journals were classified as Q1 or Q2 according to Journal Citation Report 2020 criteria. Half of the top 10 journals derived from the US (Table 4).

Figure 5 shows a dual-map overlay of academic journals, with citation relationships between citing journals (left half of the map) and cited journals (right half of the map). In Figure 5, the labels indicate the fields covered by the journal, the colored lines depict different citation paths, and the path width is proportional to the z-score scale. We noted two main citation paths, the orange path and the green path, which indicated that articles from molecular, biology, and genetics journals were frequently cited by articles from molecular, biology, immunology, medicine, medical, and clinical journals.

### 3.5. Analysis of Co-Cited References

Table 5 shows the top ten co-cited references on pancreatic cancer immunotherapy. “Safety and activity of anti-PD-L1 antibody in patients with advanced cancer” by Brahmer et al. [13] was the highest cited article with 387 citations. This article mainly found that antibody-mediated PD-L1 blockade induced durable tumor regression and long-term disease stabilization in patients with advanced cancers. The second and third articles were by Royal et al. [28] and Conroy et al. [29] and had 319 and 291 citations, respectively.

We also performed a timeline analysis of co-cited references to reveal the changes in research hotspots over time (Figure 6A). The vertical axis in Figure 6A represents the clustering label, with a total of 13 clusters, and the horizontal axis shows the timing of the occurrence of important reference nodes. “#3 carcinoembryonic antigen” and “#11 common fragile site” were the earliest hotspots, and there were relatively few studies on molecular therapy in recent years. The most recent research hotspots were “#6 prognostic biomarker” and “#5 csf-secreting vaccine”, suggesting that concerns are shifting more toward cancer genes and immunotherapy research. In the co-citation network analysis of the references, the references were divided into 10 clusters, the weighted mean silhouette and modularity Q were both greater than 0.8, and the clustering structure was reasonable. The top three clusters were “#0 molecular therapy”, “#1 biological approaches”, and “#2 future prospect” (Figure 6B).

“Citation bursts” refer to references well known in the field and widely cited during a particular period. CiteSpace identified 15 most frequently cited references (Figure 7). In Figure 7, the red segmented lines indicate the active period. A short citation burst in the field of pancreatic cancer immunotherapy began in 2011, corresponding to the advent of immunotherapy for pancreatic cancer. Other bursts emerged later, and have developed relatively quickly in recent years. The most recent highly cited references (2019–2021) were Rahib et al. [30], Zhu et al. [31], and Le et al. [32], and these bursts are still ongoing.

### 3.6. Keywords Analysis of Research Hotspots

We used VOSviewer to create a keyword network map, overlay map, and density map. There were a total of 7815 keywords in 2475 articles, of which 223 keywords occurred 20 or more times and 30 keywords occurred more than 100 times. Keywords that appeared at least 20 times were analyzed. 

In the keyword co-occurrence analysis, we found that the last 15 years of pancreatic cancer immunotherapy research can be summed up in the following four main topics that had the highest number of publications: “immunotherapy”, “clinical treatment study”, “tumor immune cell expression”, and “tumor microenvironment”. In addition, each topic covered many small branches, as evident from the colors of each main topic indicated by large circles and related smaller circles of the same color. In Figure 8A, the main blue-marked cluster “immunotherapy” can be subdivided into the branches “t-cell immunity”, “PD-L1”, “nivolumab”, and “pembrolizumab”. The green cluster is related to “clinical treatment”, focusing on the clinical efficacy of conventional treatment, with keywords such as “gemcitabine”, “radiation therapy”, “adjuvant therapy”, and “targeted therapy”. Table 6 reported the current ongoing trials on immunotherapy for pancreatic cancer. The yellow cluster is mainly related to anti-tumor effects of immune cells and biological effect of immunotherapy, which may aid in elucidating the mechanisms of pancreatic cancer immunity. The red cluster is related to the “tumor microenvironment”, with keywords such as “tumor-associated macrophage”, “suppressor T cell”, “muc-1”, and “regulatory T-cell”.

In Figure 8B, keywords are colored according to the average year of their occurrence in an overlay map. In the past decade, “nivolumab”, “vaccine”, and “gemcitabine” appeared frequently, suggesting that immune clinical trials were initiated and traditional anti-cancer therapies were well researched. The tumor microenvironment and immune checkpoint inhibitors have been studied more recently, and may be the hotspots of future research.

Another method for identifying research hotspots is to focus on keywords with strong burst strength (Figure 9A). From 2007 to 2015, studies focused on anti-tumor immune mechanisms, as indicated by the high burst strength of keywords such as “dendritic cell”, “colony stimulating factor”, “lymphocyte”, and “regulatory T cell”, etc. From 2015 to 2021, keywords such as “vaccine” and “immune checkpoint inhibitors” have high burst strength, suggesting that these areas are becoming new research hotspots. Immunotherapy and pancreatic cancer were the most frequently occurring keywords, consistent with the objectives of this study (Figure 9B).

## 4. Discussion

We live in an age of information explosion; the speed of information generation is accelerating worldwide [33]. In the past 10 years, big data has become one of the most commonly used terms in healthcare, education, and industrial sectors [34,35]. Medical big data and medical data mining are multidisciplinary research fields, combining medicine and computer science, and have received wide attention, with the rapid development of information technology [34,36]. In this study, VOSviewer and CiteSpace software were used to analyze 15 years of research in the field of pancreatic cancer immunotherapy. We systematically reviewed the development of trends in the field by analyzing the core authors, highly productive countries and institutions, core journals, and keyword clusters.

Over the past 15 years, the literature on pancreatic cancer immunotherapy has been increasing (Figure 2A). Thus, it seems safe to consider this area as entering a golden age in the coming years. The US had the highest productivity. However, in recent years, there has been a gradual increase in the number of publications from China, Germany, and Japan, indicating a growing interest in immunotherapy for pancreatic cancer in these countries (Figure 2B,C). Thus, we boldly predict that more publications on immunotherapy for pancreatic cancer will appear in the future as a result of growing concern.

The H-index was proposed by Ali in 2005 to evaluate the quantity and quality of the academic output of researchers, and measure the impact of their scientific contributions [37]. “Total citations” refers to the number of times that an article has been cited in other publications (e.g., articles, reports, etc.). The present analysis found that the US ranked first in influence (Table 1 and Table 2); thus, we concluded that the US is currently dominating this field. A late start in tumor immunology research may be the reason why China has lagged behind the US in this field. A co-citation analysis can depict the knowledge structure of the research area, identify research trends in the field, uncover cutting-edge research, and highlight high-impact discoveries [38]. DT Le was one of the most highly cited authors (Table 3), suggesting that this author wrote articles with greater academic impact. DT Le mainly focused on various immunotherapies, combining immunotherapy with other treatments, and exploring new treatments for tumors [39,40,41,42].

The number of citations is a good indicator for evaluating an author’s academic influence [43]. It can be seen in Table 5 that the 10 most cited studies were published between 2007 and 2021, and the most cited article was by JR Brahmer; the most striking result of this article was the confirmation of a prominent role for anti-PD-L1 in many advanced tumors [44]. Figure 6A shows a timeline view of co-cited references, which revealed developmental trends and changes in clustering across different periods. “#6 prognostic biomarker” is currently the most popular research topic.

In bibliometrics, keywords are used to understand how a field has developed. In Figure 8A, we noted that keywords related to pancreatic cancer immunotherapy were clustered into four groups. Figure 8B shows the time and frequency of the occurrence of keywords; early studies were focused on the immune system and its components (e.g., “regulatory T cell”, “dendritic cells”), and adaptive immunotherapy (e.g., “vaccine”). With the gradual deepening of immunization research, the immune microenvironment and clinical therapies are increasingly studied, and immunotherapy is attracting the attention of many researchers.

CiteSpace was used to find keywords with citation bursts, in an effort to reveal future trends in the field of pancreatic cancer immunotherapy. According to Figure 8B and Figure 9A, the colors of all keywords were divided by VOSviewer according to the average publication year (APY). The latest keyword was “immune checkpoint inhibitors” (cluster 3, APY: 2019.91), followed by “mismath repair deficiency” (cluster 3, APY: 2019.84) and “tumor microenvironment” (cluster 1, APY: 2019.27). Furthermore, “biockade” (cluster 3, APY: 2018.54), and “tumor-associated macrophages” (cluster 1, APY: 2018.70) were the recent major topics in this field. Compared with Figure 8B and Figure 9A, the study of immune checkpoint blockade was the latest, suggesting that clinical and mechanistic studies of pancreatic cancer immunity are developing together.

Furthermore, based on our analysis of keywords with the latest APY, “tumor microenvironment”, “adaptive immunotherapy”, “immunotherapy combinations”, and “molecular and gene therapy” may become research hotspots in the coming years.

Pancreatic cancer tumors are composed of malignant cells and extracellular matrix. Malignant cells only account for a small portion of tumor components; the rest is composed of fibroblasts, the extracellular matrix, endothelial cells, and hematopoietic cells. Additionally, most of the immune microenvironment of pancreatic cancer is composed of these cells [45]. Thus, the structure, as a physical barrier, protects pancreatic cancer cells against an effective delivery of chemotherapeutic agents [46]. The rapid progress in genomic analysis has led to the discovery of several FDAC susceptibility genes, such as *BRCA2*, *PALB2*, *ATM,* and K-Ras. NCCN guidelines recommend genomic analysis for all patients with PDAC. The presence of a BRCA mutation in PDAC patients may have benefits in terms of optimized treatment and longer outcome. Carcinogenesis of pancreatic cancer involves progressive accumulation of driver mutations, including the oncogene K-Ras [47] and tumour suppressor gene TP53 [48]. K-Ras mutations are not only responsible for pancreatic cancer development, but also immediately track with other mutations, contributing to the aggressiveness of pancreatic tumors [49]. To have constant proliferation and survival, pancreatic cancer cells need continuous K-Ras signaling, and numerous downstream effectors are engaged via K-Ras signaling. Accordingly, targeting K-Ras is likely to have a profound effect on pancreatic cancer [50].

The morphological evolution of pancreatic cancer begins with the formation of precursor lesions, termed pancreatic intraepithelial neoplasia [51], with increasing histological grades followed by progression to invasive adenocarcinoma. As the cancer develops, it leads to changes in the surrounding tissue stroma. A key function of any non-transformed tissue stroma is to provide homeostatic response to injury with its immune, vascular, and connective tissue components. However, cancer hijacks such physiological responses to create a favorable tumour microenvironment (TME) for its successful growth [52]. Over the past 10–15 years, numerous clinical and preclinical studies have provided abundant evidence that acellular and cellular components of the TME promote pancreatic tumorigenesis [53]. TME plays an important role in tumor development, metastasis, and resistance to chemotherapy [54], TME components can also result in the formation of connective tissue in primary and metastatic sites, or the promotion of the metastatic ability of pancreatic cancer by augmenting epithelial–mesenchymal transition (EMT) and angiogenesis [55]. TME is receiving increasing attention as a target for tumor therapy [56]. Targeting immunosuppressive cells to modulate the immune TME is a current and future focal point of research interest.

In recent years, adaptive immunotherapy has dominated cancer immunology research through literature analysis [57]. The most popular of the adaptive immunotherapies are the immune checkpoint inhibitors, CAR-T, and cancer vaccines. Cancer vaccines stimulate the presentation of immunogenic cancer antigens to the immune system, leading to activation of cancer antigen-specific CTLs in vivo [58] and subsequent anti-cancer immune response. Immune checkpoints are gaining attention as attractive targets for immunotherapy; immune checkpoints are a regulatory mechanism used to modulate the T-cell immune response [59]. Anti-CTLA 4 and anti-PD-1/PD-L1 can block immune checkpoints and activate T cells, which is an important pathway for tumor immunotherapy [17]. Checkpoint inhibitors are most successful in tumor types with high PD-L1 expression and/or high microsatellite instability or mismatch repair deficiency [39,60]. Two FDA-approved monoclonal antibodies, pembrolizumab (KEYTRUDA) and nivolumab (OPDIVO), have been developed to block the interaction between PD-1 and its ligand. Two anti-PD-L1 antibodies, atezolizumab and avelumab, have also received FDA approval. Predictably, significant resources and efforts will be devoted to the continued development of adaptive immunotherapies in the future.

Chemotherapy alone has a low median survival and is more toxic. Additionally, the large number of immunosuppressive signals and effective immune evasion found in pancreatic cancer TME and low constitutive checkpoint expression, for patients with advanced pancreatic cancer, these drugs have minimal single agent activity [61]. Many clinical studies have shown the ineffectiveness of immune checkpoint inhibitors alone [28,62]. In Table 6, clinical trials of immunotherapeutic combinations for pancreatic cancer currently under investigation are reported, combination therapy has become a growing area of PDA research, and immunotherapy combinations will become a hot spot for future research. Pembrolizumab is a humanized monoclonal antibody that targets PD-1, thereby inhibiting the interactions of PD-1 with its ligands PD-L1 and PD-L2. Therefore, it is most often used in pancreatic cancer. Gemcitabine and paclitaxel, standard first-line chemotherapy regimens for metastatic pancreatic adenocarcinoma, have been previously reported to possess immunomodulatory properties [63]. Rational immunotherapeutic combinations may prove to be the optimal way to synergistically overcome the immunosuppressive TME and shift the balance to an anti-tumor TME.

According to the keyword cluster analysis, it can be seen that molecular and gene therapy has become a research hotspot in recent years. Molecular profiling of cancer has identified potentially actionable drug targets, which has prompted attempts to discover clinically validated biomarkers to guide treatment decisions and enrollment in clinical trials [64]. Because all human cancers are primarily genetic diseases, identifying additional genes and signaling pathways could guide future research on this disease. The need to identify biomarkers that predict disease recurrence and druggable targets has driven our insights into the molecular structure of pancreatic cancer over the past 20 years of research. Despite many obstacles, the practice of guiding cancer treatment based on molecular aberrations is gaining momentum in oncology and has shown the potential to improve patient prognosis [55]. Jones S et al. examined the genetic makeup of human pancreatic cancer and identified KRAS, SMAD4, TP53, and CDKN2A as the four most frequently altered genes [65]. The research, to date, provides strong rationale for the integration of molecular profiling into clinical practice in the management of patients with pancreatic adenocarcinoma, to optimally plan a personalized therapeutic strategy through referral to specific studies of investigational targeted therapies or immunotherapies or to guide the optimal selection of standard cytotoxic chemotherapies [64].

## 5. Strengths and Limitations

This study was the first to use a bibliometric approach to analyze global research trends in pancreatic cancer immunity over a 15-year period, which can aid scholars interested in pancreatic cancer immunity in establishing a clear framework of the existing research in the field and gaining insight into its development. Secondly, the clustering analysis of high-frequency keywords performed in this study can aid scholars in understanding the hotspots and focus of the field (tumor microenvironment, adoptive immunotherapy, etc.), and provide a reference for scholars in selecting topics.

However, this study also had some limitations. Firstly, bibliometric software has high standards and specifications for the data. Therefore, to ensure the quality and integrity of the collected data, we only searched the WoSCC database. The lack of a complement from other literature databases (such as Scopus), may have led to an incomplete analysis of the data. Secondly, this study only analyzed basic publication information and lacked in-depth exploration of the specific content, which is somewhat subjective. Finally, we only selected articles published in English; thus, significant articles published in other languages may have been excluded.

## 6. Conclusions

This study analyzed 15 years of immune-related research on pancreatic cancer using VOSviewer and Citespace software, and systematically reviewed the trends in the field. The collaborative community of authors in this field is still in the process of formation, but there are already several well-known authors. The core journals that publish papers in this field are Cancers and Clinical Cancer Research. Scholars in the US publish the most articles, accounting for 44% of the articles, and are highly recognized in the field, in terms of the average number of citations per article. Keyword co-occurrence and cluster analyses revealed that several stable research themes have developed in this research field, such as the clinical treatment of pancreatic cancer, adoptive immunotherapy, etc. The author co-citation analysis revealed that the research hotspots in this field are constantly changing, and a careful analysis of the literature of highly cited authors may further explain the changing trend in hotspots.

## Figures and Tables

**Figure 1 healthcare-11-00304-f001:**
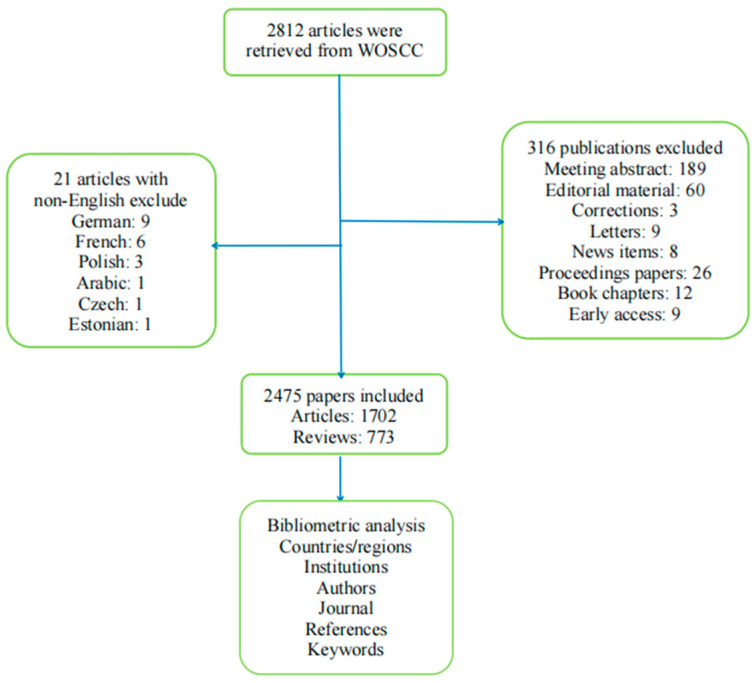
Schematic diagram of the search process.

**Figure 2 healthcare-11-00304-f002:**
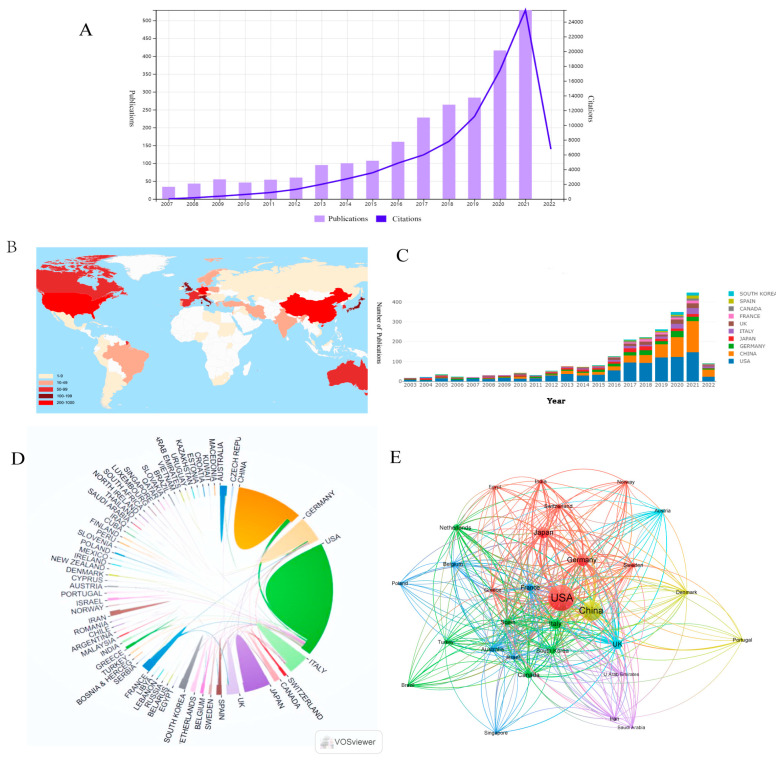
(**A**) The annual number and total citations of articles related to pancreatic cancer immunotherapy published from 2007 to 2021. (**B**) World map based on the number of articles of countries/regions. (**C**) The trend of annual number of articles from top ten countries/regions. (**D**) The international collaborations between countries/regions. The links between countries/regions represented the frequency of the collaborations. (**E**) The citation network between the top 30 countries/regions in terms of article quantity, generated by VOSviewer. The size of nodes indicates the number of articles, while the thickness of links indicates the citation strength.

**Figure 3 healthcare-11-00304-f003:**
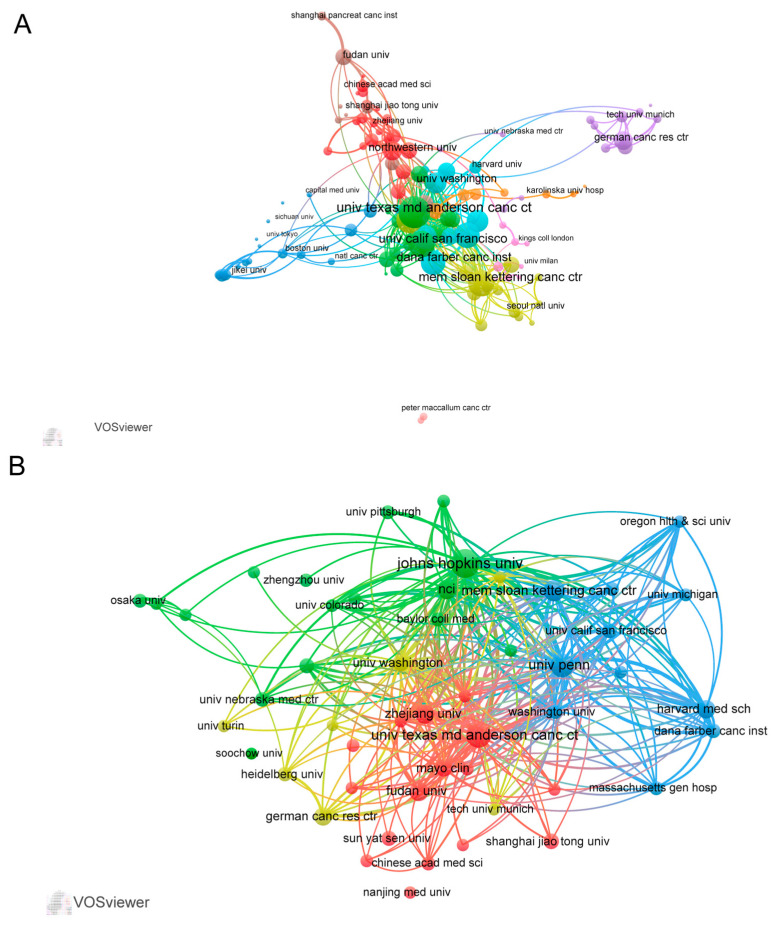
(**A**) The cooperation map of institutions performed with VOS viewer. The size of nodes represented the number of publications, while the thickness of the lines indicated the collaboration strength. (**B**) The citation network of institutions created with VOS viewer.

**Figure 4 healthcare-11-00304-f004:**
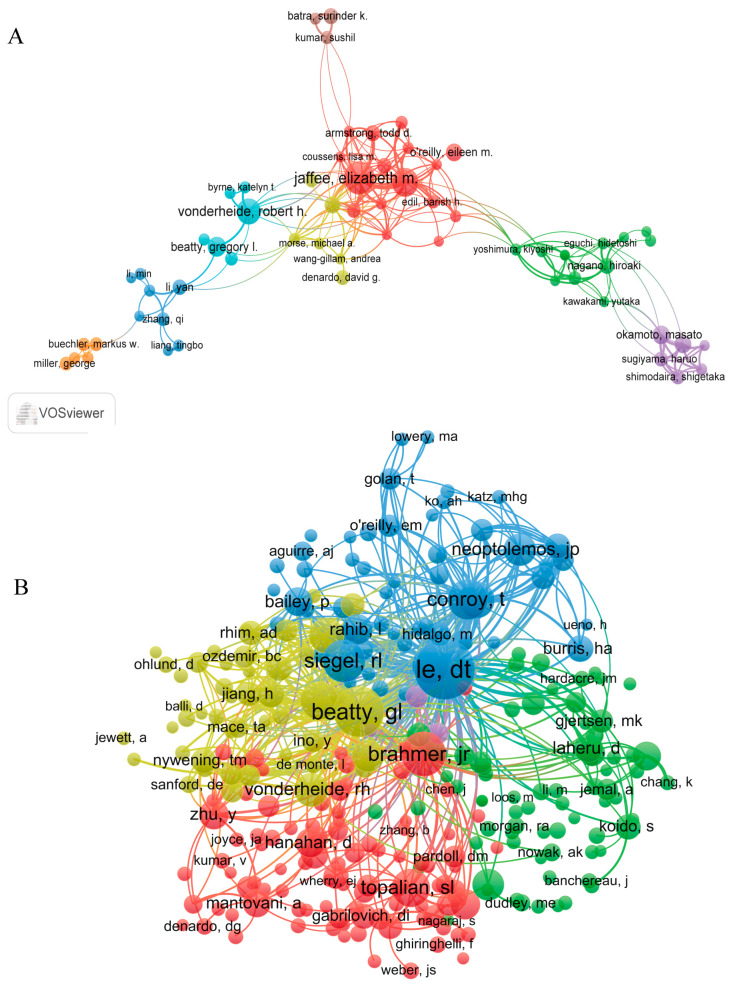
The cooperation map of authors (**A**) and co-citation map of authors (**B**) conducted with VOSviewer.

**Figure 5 healthcare-11-00304-f005:**
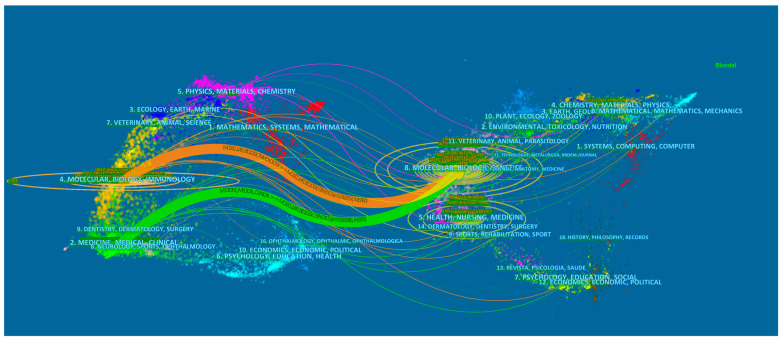
A dual-map overlap of journals on immunotherapy for pancreatic cancer completed by CiteSpace.

**Figure 6 healthcare-11-00304-f006:**
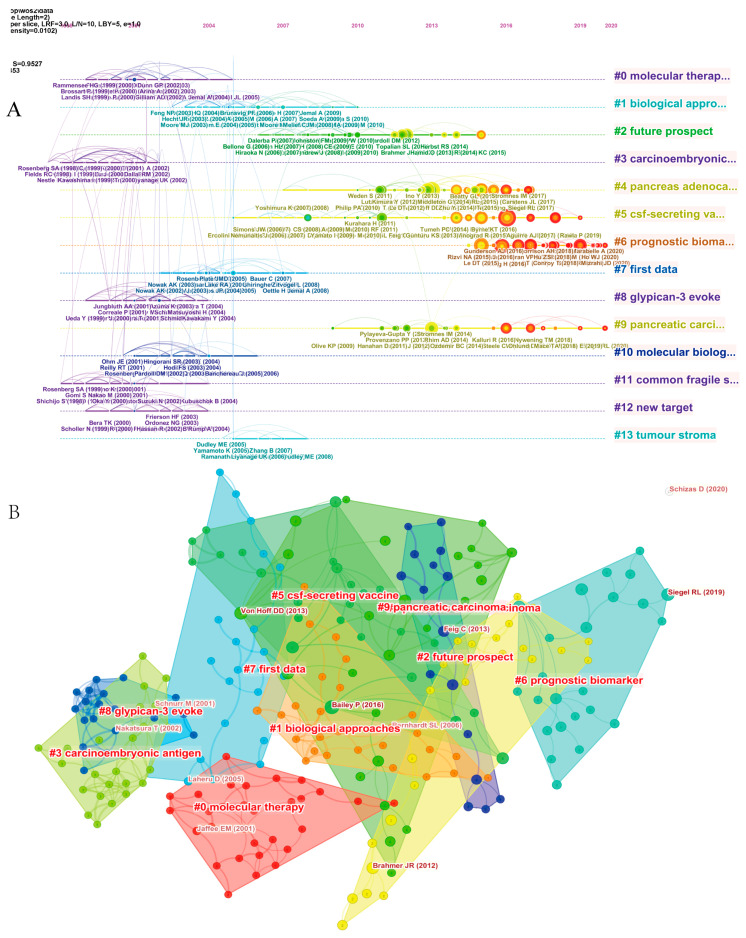
Timeline view (**A**) and cluster view (**B**) of co-citation references related to pancreatic cancer immunotherapy visualized by CiteSpace. Fifteen labeled clusters are colored on the right. The nodes on the line represented the cited references.

**Figure 7 healthcare-11-00304-f007:**
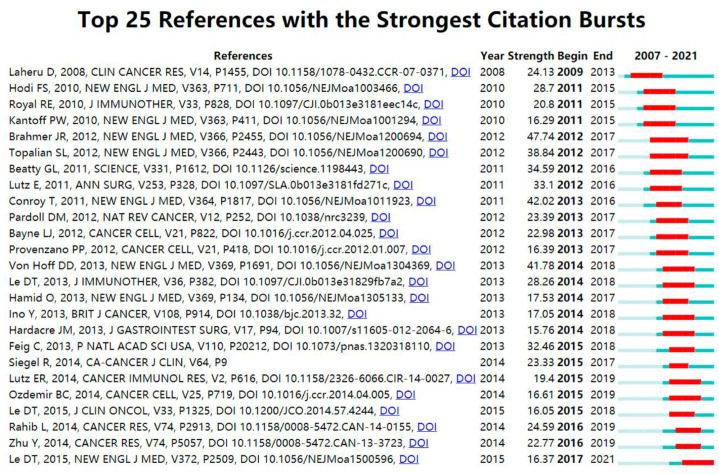
The top 25 references with the strongest citation bursts during 2007 to 2021.

**Figure 8 healthcare-11-00304-f008:**
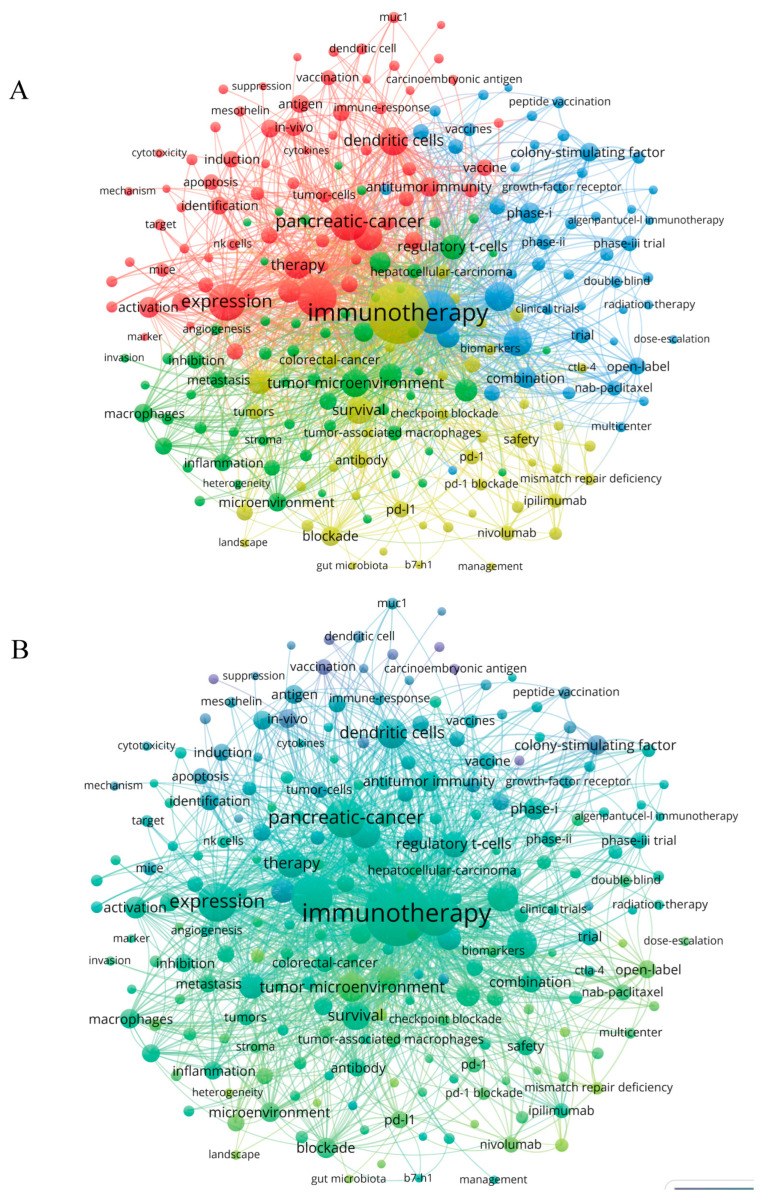
(**A**) The network of 223 keywords with a frequency of no fewer than 20 times. All keywords could be grouped into four clusters. (**B**) The overlay visualization map showed that keywords are colored according to their average occurrence time.

**Figure 9 healthcare-11-00304-f009:**
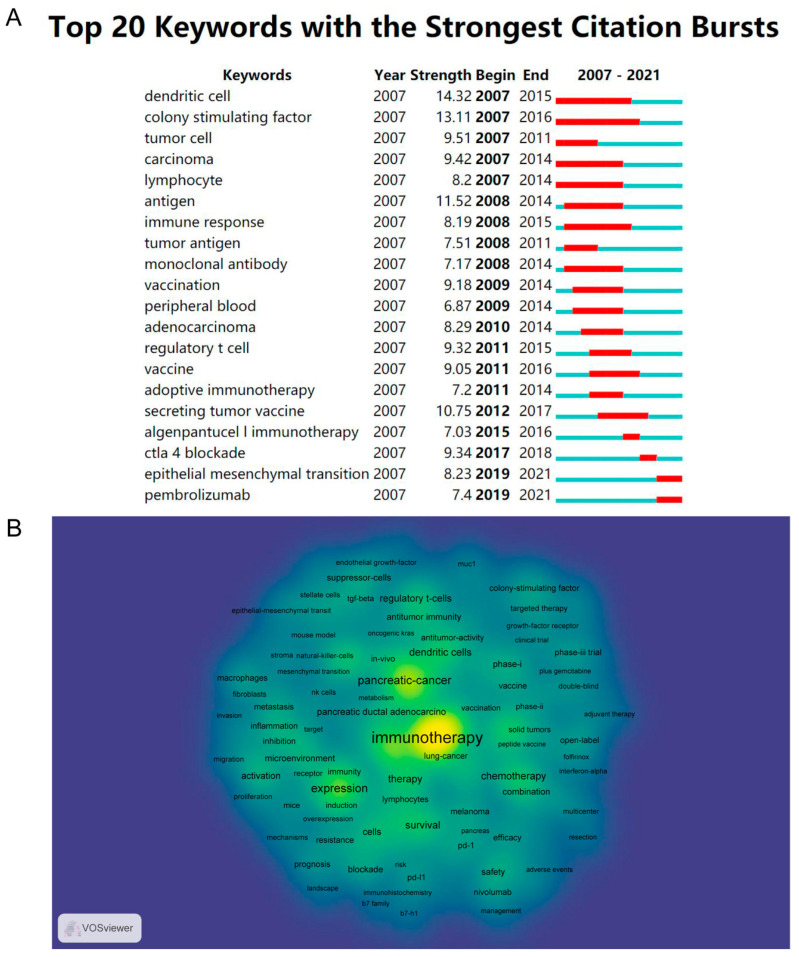
(**A**) Top 20 keywords with the strong citation bursts on immunotherapy for pancreatic cancer (**B**) density map of keywords generated by the VOS viewer.

**Table 1 healthcare-11-00304-t001:** Top 10 productive countries/regions.

Rank	Country	Counts	Citations	H-Index	Percentage	Average Citation Per Paper
1	USA	1096	57,699	111	44.282	52.65
2	China	567	10,565	50	22.91	18.63
3	Germany	222	8292	46	8.97	37.35
4	Japan	198	6745	44	8.00	33.97
5	Italy	156	6760	37	6.30	43.33
6	UK	133	6865	38	5.37	51.62
7	France	89	3901	30	3.60	43.83
8	Canada	70	3005	25	2.82	42.93
9	South Korea	67	2358	22	2.70	35.19
10	Spain	58	2161	22	2.34	37.26

**Table 2 healthcare-11-00304-t002:** Top 10 productive institutes.

Rank	Institutions	Country	TLS (Total Link Strength)	Count	Citations
1	Johns Hopkins University	USA	2313	103	6929
2	University Texas, MD Anderson Cancer center	USA	1533	80	10,901
3	University of Pennsylvania	USA	2384	70	8136
4	Memorial Sloan-Kettering Cancer	USA	1187	59	4031
5	NCI	USA	913	50	4906
6	Fudan University	China	633	49	944
7	Zhejiang University	China	538	48	1094
8	German Cancer Research Center	Germany	518	44	2319
9	Harvard Medical School	USA	678	42	2778
10	University of Washington	USA	1437	37	8128

TLS: total link strength (TLS).

**Table 3 healthcare-11-00304-t003:** The top ten authors and co-cited authors.

Rank	Author	Documents	Citations	Co-Cited Author	Citations	TLS
1	Jaffee, E.M.	43	3598	Le, D.T.	749	20,853
2	Zheng, L.	34	2095	Beatty, G.L.	550	17,327
3	Vonderheide, R. H.	29	5573	Brahmer, J.R.	451	10,346
4	Beatty, G. L.	18	2487	Siegel, R.L.	414	6779
5	Novelli, F	18	709	Conroy, T	379	9157
6	Cappello, P	17	683	Topalian, S.L.	347	6582
7	Okamoto, M	17	349	Feig, C	339	9204
8	Koido, S	16	348	Royal, R.E.	324	8797
9	Oreilly, E.M.	15	544	Vonhoff, D.D.	324	7848
10	Le, D.T.	14	2150	Hingorani, S.R	301	7750

TLS: total link strength (TLS).

**Table 4 healthcare-11-00304-t004:** The top ten most active journals.

Rank	Journal title	Country	Documents	Percentage (N/2475)	IF (2020)	Citations	JCR (2020)
1	Cancers	Switzerland	99	4.0	6.639	1119	Q2/3
2	Clinical Cancer Research	USA	88	3.56	12.53	6490	Q1
3	Oncoimmunology	USA	69	2.79	8.11	1889	Q1
4	Frontiers in Immunology	Switzerland	68	2.74	7.56	1259	Q2
5	Journal for Immunotherapy of cancer	UK	55	2.22	13.75	1321	Q1
6	Cancer Immunology Immunotherapy	USA	52	2.10	6.968	1455	Q2
7	Frontiers in Oncology	Switzerland	43	1.73	6.244	483	Q2
8	Cancer Immunology Research	USA	41	1.66	11.151	2125	Q2
9	International Journal of Molecular Sciences	Switzerland	38	1.54	5.924	722	Q1/2
10	Cancer Research	USA	36	1.46	12.701	2878	Q1

**Table 5 healthcare-11-00304-t005:** Top 10 co-cited references concerning the research.

Title	First Author	Journal	Year	Type	Citations
Safety and activity of anti-PD-L1 antibody in patients with advanced cancer	Julie R Brahmer	The New England Journal Of Medicine	2012	Article	387
Phase 2 trial of single agent Ipilimumab (anti-CTLA-4) for locally advanced or metastatic pancreatic adenocarcinoma	Richard E Royal	Journal of immunotherapy	2010	Article	319
FOLFIRINOX versus gemcitabine for metastatic pancreatic cancer	Thierry Conroy	The New England Journal Of Medicine	2011	Article	291
Targeting CXCL12 from FAP-expressing carcinoma-associated fibroblasts synergizes with anti-PD-L1 immunotherapy in pancreatic cancer	Christine Feig	PNAS	2013	Article	234
Increased Survival in Pancreatic Cancer with nab-Paclitaxel plus Gemcitabine	Daniel D. Von Hoff	The New England Journal Of Medicine	2013	Article	265
CD40 agonists alter tumor stroma and show efficacy against pancreatic carcinoma in mice and humans	Gregory L Beatty	Science	2011	Article	205
Evaluation of ipilimumab in combination with allogeneic pancreatic tumor cells transfected with a GM-CSF gene in previously treated pancreatic cancer	Dung T Le	Journal of immunotherapy	2013	Article	165
Projecting cancer incidence and deaths to 2030: the unexpected burden of thyroid, liver, and pancreas cancers in the United States	Lola Rahib	Cancer Research	2014	Article	220
Dynamics of the immune reaction to pancreatic cancer from inception to invasion	Carolyn E Clark	Cancer Research	2007	Article	200
Genomic analyses identify molecular subtypes of pancreatic cancer	Bailey, Peter	Nature	2016	Article	171

**Table 6 healthcare-11-00304-t006:** Ongoing clinical trials investigating immunotherapy in pancreatic cancer.

Study	Disease	Phase	ID
Sindilizumab in combination with albumin-bound paclitaxel and gemcitabine	Resectable and junctionally resectable pancreatic cancer	II	ChiCTR2200064424
Recombinant human endostatin combined with PD1 inhibitor and chemotherapy	pancreatic cancer	IV	ChiCTR2200064322
Surufatinib plus Envafolimab plus Tegafur-Gimeracil-Oteracil Potassium Capsule	Advanced pancreatic cancer	II	ChiCTR2200060760
Carrelizumab, albumin paclitaxel and apatinib combined with or without Fuzi Lizhong Pill	Unresectable locally advanced/metastatic pancreatic cancer	II	ChiCTR2200058126
T Cell Receptor Modified T (TCR-T) Cells	Advanced Pancreatic Cancer, Lung Cancer and Colorectal Cancer	0	ChiCTR2200057171
Chidamide in combination with anti-PD-(L)1 and gemcitabine-based chemotherapy	Advanced pancreatic cancer	II	ChiCTR2200055523
Nituzumab combined with albumin paclitaxel combined with teggio	Advanced or metastatic pancreatic cancer	0	ChiCTR2100051515
Gemcitabine combined with Regorafenib and GM-CSF combined with Sintilimab combined with or without IBI310	Advanced pancreatic cancer	II	ChiCTR2100049459
Camrelizumab combined with albumin-bound paclitaxel and apatinib mesylate	Unresectable pancreatic cancer	IV	ChiCTR2100045844

## Data Availability

Not applicable.

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
