# Peer review of "Bibliometric Analysis of Hotspots and Frontiers of Immunotherapy in Pancreatic Cancer"

_healthcare, 2023, doi:10.3390/healthcare11030304_

Round 1

Reviewer 1 Report

In this manuscript, the authors evaluated the research trends in pancreatic cancer immunotherapy, analyzing 15 years of immune-related research. The paper is interesting, the methods are conform. For publication, the manuscript needs some modifications.

Introduction

The authors should include a section on hereditary pancreatic cancer, mutations in susceptibilty genes and therapy in mutated patients,  also since they refer to in the 'discussion' section. In this regard, the authors could benefit from reading the following article, the contents of which could be useful for improving the manuscript:

     PMID: 35205366    DOI: 10.3390/genes13020321

     PMID: 36290818  DOI: 10.3390/curroncol29100541

Results

- In Figure 1 the excluded articles are 22 and not 21, please correct;

it would be better to also indicate in the text the 316 excluded publications, and not only in Figure 1.

- Table 2 and 3

Please indicate the meaning of 'TLS'.

- The reference to table 4 does not appear in the text.

Discussion

The authors refer to Big Data, it would be preferable to add a few sentences in the ‘introduction’ section explaining  what is meant by Big Data, for greater completeness of the manuscript.

Author Response

Dear Editor:

I am submitting a revision of the above referenced manuscript by Xu et al, entitled “Bibliometric Analysis of Hotspots and Frontiers of Immunotherapy in Pancreatic Cancer” (ID: healthcare-2069959), for consideration for publication in Healthcare.

Thank you very much for your email on review comments. We appreciate your patience and valuable suggestions on our manuscript. We have carefully revised the manuscript according to the suggestions and guidelines. The revised manuscript has been uploaded to Healthcare online manuscript submission and tracking system. In this manuscript, the major modifications were noted with red font. According to the comments of reviewers, we will answer point-by-point response in the file of “Point-by-point response to reviewers”.

We thank the reviewers for their insightful comments and helpful suggestions that have helped us to strengthen the manuscript substantially. We believe that we have addressed the reviewers’ concerns fully and to the best of our ability. We hope the paper now is fit for publication in Healthcare.

Thank you very much for your consideration. I look forward to hearing from you favorably.

Best regards,

Project team members

Reviewer 1:

The manuscript has been improved according to the suggestions of reviewers:

Reviewers' comments:

Introduction

The authors should include a section on hereditary pancreatic cancer, mutations in susceptibilty genes and therapy in mutated patients,  also since they refer to in the 'discussion' section. In this regard, the authors could benefit from reading the following article, the contents of which could be useful for improving the manuscript:

     PMID: 35205366    DOI: 10.3390/genes13020321

     PMID: 36290818  DOI: 10.3390/curroncol29100541

Results

- In Figure 1 the excluded articles are 22 and not 21, please correct;

it would be better to also indicate in the text the 316 excluded publications, and not only in Figure 1.

- Table 2 and 3

Please indicate the meaning of 'TLS'.

- The reference to table 4 does not appear in the text. 

Discussion

The authors refer to Big Data, it would be preferable to add a few sentences in the ‘introduction’ section explaining  what is meant by Big Data, for greater completeness of the manuscript.

Reply:

  • As you said“The authors should include a section on hereditary pancreatic cancer, mutations in susceptibilty genes and therapy in mutated patients,  also since they refer to in the 'discussion' section.” 

Response: We sincerely thank the reviewer’s helpful suggestions. We are particularly grateful for the references you have provided. We have carefully read the article, we have gained a lot of. We do very much agree with you. Most of cases of pancreatic cancer are sporadic, however about 5% to 10% report a hereditary predisposition. Thus, hereditary pancreatic cancer should be mentioned in the citation. Hereditary pancreatic cancer has been added to this article. The following description has added into the revised manuscript.

 Page 1-2 Line 43-51:“ It is reported that about 5% ~ 10% of pancreatic cancers are caused by genetic factors (1). NCCN guidelines suggested the genetic analysis of mutation for PDAC pathogenesis-associated syndromes. There are many types of genetic mutations associated with hereditary pancreatic cancer, including ATM, BRCA1, BRCA2, CDKN2A, PALB2, STK11, MLH, MSH2, MSH6 and PMS2. Among them, BRCA2 mutations are the most common genetic alteration, observed in about 5-10% of cases (2). For the treatment of patients with BRCA mutations, NCCN guidelines recommend the combination of gemcitabine (GEM) and cisplatin (CDDP). PARP inhibitors have been clinically investigated for BRCA mutated cases (3).”

Page 1-2 Line 55-57:“A clinical trial of anti-programmed cell death protein 1 (PD-1) antibody reported a positive response in PDAC patients with MMR alterations. Thus, among those strategies, immunotherapy may be one of the most promising.(4)”

Page17 Line 384-388:“The rapid progress in genomic analysis has led to the discovery of several FDAC susceptibility genes, such as BRCA2, PALB2, ATM and K-Ras. NCCN guidelines recommend genomic analysis for all patients with PDAC. The presence of a BRCA mutation in PDAC patients may have benefits in terms of optimized treatment and longer outcome.”

  1. Pilarski R. The Role of BRCA Testing in Hereditary Pancreatic and Prostate Cancer Families. Am Soc Clin Oncol Educ Book(2019) 39: 79-86 .doi:10.1200/EDBK_238977
  2. Lai E, Ziranu P, Spanu D, Dubois M, Pretta A, Tolu Set al. BRCA-mutant pancreatic ductal adenocarcinoma. BRIT J CANCER(2021) 125: 1321-1332 .doi:10.1038/s41416-021-01469-9
  3.   NCCN Guidelines Version 2. 2018 Pancreatic Adenocarcinoma. [(accessed on 25 December 2018)]. Available online: https://www.nccn.org/professionals/physician_gls/default.aspx
  4. 4. Ohmoto A, Yachida S, Morizane C. Genomic Features and Clinical Management of Patients with Hereditary Pancreatic  Cancer Syndromes and Familial Pancreatic Cancer. INT J MOL SCI(2019) 20 .doi:10.3390/ijms20030561

  • For the secondquestion that “In Figure 1 the excluded articles are 22 and not 21, please correct.”

Response: We apologize that we have incorrectly marked“22” as “21”in figure 1, We have verified the original references one by one, We found that the excluded articles you mentioned is indeed 22 and not 21, we have corrected and removed irrelevant reference. Thanks very much for your attention to our paper.

  • For the third question that“it would be better to also indicate in the text the 316 excluded publications, and not only in Figure 1.

- Table 2 and 3

Please indicate the meaning of 'TLS'.

- The reference to table 4 does not appear in the text.” 

Response: We sincerely thank the reviewer’s insightful comments and helpful suggestions that have helped us to strengthen the manuscript substantially. We have made corrections based on your suggestions. We have added a section on Article Selection, in which 316 excluded documents are mentioned, including Meeting Abstracts, Editorial Materials, Corrections, Letters, News Items, Proceedings Papers,  Book Chapters and Early Access.

Page 3 Line 117-126:“ 2.2. Article Selection  The researchers screened all articles by the title and abstract and selected the article that fits the topic. Data were downloaded and analyzed independently by two researchers to ensure its accuracy and reliability. When the two researchers had different opinions regarding the data, a third researcher decided the prevailing opinion. The exclusion criteria were document types (Meeting Abstracts or Editorial Materials or Corrections or Letters or News Items or Proceedings Papers or Book Chapters or Early Access). The publications from January 2007 to December 2021 were searched. The language was limited to English, excluding German, French, Polish, Arabic, Czech and Estonian. The literature types were limited to “articles” and “review articles”.”

We explained the meaning of TLS in Tables 2 and 3, where TLS means total link strength.

As you said “The reference to table 4 does not appear in the text.” This mistake has been corrected. We will be happy to edit the text further based on helpful comments from the reviewers.

  • For the fourth question that:“The authors refer to Big Data, it would be preferable to add a few sentences in the ‘introduction’ section explaining what is meant by Big Data, for greater completeness of the manuscript.”

Response: We appreciate for your suggestions. We do very much agree with you. As you can see, We have added the big data content in the Introduction section to make the article more coherent.

The manuscript was modified as follows:

Page 2 Line 85-89:“ Medical big data and data mining is the analysis of a large amount of medical data to dig out valuable diagnostic rules to provide reference for the diagnosis and treatment of diseases. Bibliometrics is the analysis of a large amount of data to identify research hotspots and explore cutting-edge trends in recent years by revealing the structure of knowledge in a scientific field over a certain period of time.”

Reviewer 2 Report

The review is basically well-written and provides comprehensive bibliometric analyses of immunotherapy in pancreatic cancer. However, I have some comments:

1. The resolution of figures are too low and the words on the images are too small (e.g., Fig 6, Fig 8), so it is hard for readers to get the information. Please provide high-resolution images or enlarge them.  

2. On Page 15, Line 319, the authors mentioned “Table 6A”, which does not exist in the current review.

3. In the Section 4 “Discussion”, the authors firstly discussed their bibliometric analyses, followed by “4.1 Research Hot spots”, which however seems abrupt, both in logic and in content. It would be better to make it a separate section.

4. So far as I know, many clinical trials in pancreatic cancer were initiated, as the authors mentioned on Page 12, Line 266. It may be useful to provide a table of the clinical trials and their therapeutic efficacies regarding the immunotherapy in pancreatic cancer in the section of “Research Hot spots”.  

5. The overall writing is fine but somewhat Chinglish. It would be better to improve the English language and style. 

Author Response

Dear Editor:

I am submitting a revision of the above referenced manuscript by Xu et al, entitled “Bibliometric Analysis of Hotspots and Frontiers of Immunotherapy in Pancreatic Cancer” (ID: healthcare-2069959), for consideration for publication in Healthcare.

Thank you very much for your email on review comments. We appreciate your patience and valuable suggestions on our manuscript. We have carefully revised the manuscript according to the suggestions and guidelines. The revised manuscript has been uploaded to Healthcare online manuscript submission and tracking system. In this manuscript, the major modifications were noted with red font. According to the comments of reviewers, we will answer point-by-point response in the file of “Point-by-point response to reviewers”.

We thank the reviewers for their insightful comments and helpful suggestions that have helped us to strengthen the manuscript substantially. We believe that we have addressed the reviewers’ concerns fully and to the best of our ability. We hope the paper now is fit for publication in Healthcare.

Thank you very much for your consideration. I look forward to hearing from you favorably.

Best regards,

Project team members

Reviewer 2:

The review is basically well-written and provides comprehensive bibliometric analyses of immunotherapy in pancreatic cancer. However, I have some comments:

  1. The resolution of figures are too low and the words on the images are too small (e.g., Fig 6, Fig 8), so it is hard for readers to get the information. Please provide high-resolution images or enlarge them.  
  2. On Page 15, Line 319, the authors mentioned “Table 6A”, which does not exist in the current review.
  3. In the Section 4 “Discussion”, the authors firstly discussed their bibliometric analyses, followed by “4.1 Research Hot spots”, which however seems abrupt, both in logic and in content. It would be better to make it a separate section.
  4. So far as I know, many clinical trials in pancreatic cancer were initiated, as the authors mentioned on Page 12, Line 266. It may be useful to provide a table of the clinical trials and their therapeutic efficacies regarding the immunotherapy in pancreatic cancer in the section of “Research Hot spots”.  
  5. The overall writing is fine but somewhat Chinglish. It would be better to improve the English language and style. 

Reply:

We really appreciate your patience and valuable suggestions on our manuscript. 

  1. The resolution of figures are too low and the words on the images are too small (e.g., Fig 6, Fig 8), so it is hard for readers to get the information. Please provide high-resolution images or enlarge them.  

Response: We deeply appreciate your review and valuable comments above. We strongly agree with your opinion. We have checked all the images and enlarged the resolution of the images to make the information more accessible to the readers. If you have any better suggestions, please don't hesitate to let us know.

  1. On Page 15, Line 319, the authors mentioned “Table 6A”, which does not exist in the current review.

Response: We are very sorry for the mistakes in this manuscript and the inconvenience they caused in your reading. We have thoroughly checked and corrected the error labels that we found in our revised manuscript, we have corrected Table 6A to Figure 6A. Thanks so much for your useful comments.

  1. In the Section 4 “Discussion”, the authors firstly discussed their bibliometric analyses, followed by “1 Research Hot spots”, which however seems abrupt, both in logic and in content. It would be better to make it a separate section.

Response: We appreciate your valuable suggestions on our paper. I strongly agree with your comments. As you say, it seems abrupt to single out research hot spots in the discussion. We removed the separate heading Research Hotspots. We have read a large amount of bibliometrics-related literature, and we thought that research hotspots should fall under the category of discussion.

  1. So far as I know, many clinical trials in pancreatic cancer were initiated, as the authors mentioned on Page 12, Line 266. It may be useful to provide a table of the clinical trials and their therapeutic efficacies regarding the immunotherapy in pancreatic cancer in the section of “Research Hot spots”.  

Response: We appreciate your valuable suggestions on our paper. Pancreatic adenocarcinoma was the fourth leading cause of death from cancer in the United States in 2010, and it carries a grim prognosis: the 5-year survival rate is 6% in Europe and the United States (1,2), that's why pancreatic cancer research has become a hot topic. Our research group is engaged in the research on pancreatic cancer for many years. When we read the related literature, we found that interest in immunotherapy for pancreatic cancer has drastically increased in recent years and numerous related papers have been published. However, rapid growth of publications on immunotherapy for pancreatic cancer makes it difficult not only to keep track of the new developments, but also for new researchers to identify relevant information and new directions in this area. In this study, we have identify the scientific output and activity related to immunotherapy for pancreatic cancer through bibliometric approaches.

“Immunotherapy combinations” may become research hotspots in the coming years. We reviewed the data and found that a large number of clinical trials of immunotherapy combinations have been conducted, and we added the contents of Table 6 for a large number of clinical trials of immunotherapy combinations.

The manuscript was modified as follows:

Page 18 and Line 437-440:“In Table 6, clinical trials of immunotherapy combinations for pancreatic cancer currently under investigation are reported, combination therapy has become a growing area of PDA research, and immunotherapy combinations will become a hot spot for future research.”

References

  1. 1. Jemal A, Siegel R, Xu J, Ward E. Cancer statistics, 2010. CA Cancer J Clin(2010) 60: 277-300 .doi:10.3322/caac.20073
  2. 2. Sant M, Allemani C, Santaquilani M, Knijn A, Marchesi F, Capocaccia R. EUROCARE-4. Survival of cancer patients diagnosed in 1995-1999. Results and commentary. EUR J CANCER(2009) 45: 931-991 .doi:10.1016/j.ejca.2008.11.018

  1. The overall writing is fine but somewhat Chinglish. It would be better to improve the English language and style. 

Response: We apologize for the Chinglish of our manuscript. We worked on the manuscript for a long time and repeated addition and removal of sentences and sections obviously led to poor readability. We have now worked on both language and readability and have also involved native English speakers for language corrections. We really hope that the flow and language level have been substantially improved.

We sincerely thank you for their reviews and comments. We learn a lot and have rewritten the respective part according to the reviewer’s suggestions. The new manuscript has been re-submitted, in which the amended sentences are highlighted in red color. We especially like this journal and hope our revised manuscript can be accepted for publication. We will do our best to refine our manuscript.

Round 2

Reviewer 2 Report

The manuscript is much improved and most of the comments are addressed except for some grammar errors which need to be further edited.